# Direct Innominate Artery Cannulation for Thoracic Aortic Surgery

**DOI:** 10.3390/jcm14082684

**Published:** 2025-04-14

**Authors:** Corrado Cavozza, Rossella Scarongella, Giulia Policastro, Giulia Maj, Antonella Cassinari, Serena Penpa, Antonio Maconi, Andrea Audo

**Affiliations:** 1Cardiac Surgery Department, University Hospital “SS. Antonio e Biagio e C. Arrigo”, 15121 Alessandria, Italy; rossella.scarongella@ospedale.al.it (R.S.); aaudo@ospedale.al.it (A.A.); 2Cardiac Anesthesia Department, University Hospital “SS. Antonio e Biagio e C. Arrigo”, 15121 Alessandria, Italy; giulia.policastro1991@gmail.co (G.P.); giulia_maj@yahoo.it (G.M.); 3Research and Innovation Department (DAIRI), University Hospital “SS. Antonio e Biagio e C. Arrigo”, 15121 Alessandria, Italy; antonella.cassinari@ospedale.al.it (A.C.); serena.penpa@ospedale.al.it (S.P.); amaconi@ospedale.al.it (A.M.)

**Keywords:** thoracic aorta surgery, innominate artery cannulation

## Abstract

**Objectives**: Direct innominate artery cannulation is a viable and effective alternative for arterial inflow during thoracic aorta surgery, applicable in elective and emergent cases. This technique ensures reliable circulatory control. **Methods**: A single-center retrospective study of 208 cases that underwent thoracic aortic surgery between January 2010 and December 2021 was performed. The primary outcomes were in-hospital and remote mortality and the secondary outcomes were adverse neurologic events. **Results**: The median age of the patients was 69 years. The male gender accounted for 63.9% of the cases. The most represented surgical interventions consisted of hemiarch replacement in 105 cases (50.5%) and ascending aorta and aortic valve replacement (wheat procedure) in 71 cases (34.1%). The operative mortality rate was 5.3%, with six cases attributed to aortic-type dissection. The overall remote mortality rate at five years was 7.7. Postoperatively, 70 patients experienced alterations in the level of consciousness, with 12 of these cases belonging to the dissection group. Six patients with permanent neurologic symptoms had a positive computed tomography scan. Of the eleven patients with negative brain computed tomography scans, nine experienced temporary neurological deficits, while two suffered from permanent neurological damage. **Conclusions**: Direct innominate artery cannulation represents a safe and effective method for providing arterial inflow during cardiopulmonary bypass, offering an outstanding alternative to traditional sites for both planned and urgent surgical interventions.

## 1. Introduction

The surgical management of thoracic aortic pathologies is highly complex, requiring meticulous imaging evaluation, customized surgical strategies, precise perfusion management, and continuous monitoring of end-organ perfusion supply [1,2,3]. Open surgical approaches set the standard for addressing various acute and chronic thoracic aortic pathologies [3]. Selecting the optimal arterial cannulation site is a critical component of operative planning for the distal ascending aorta and aortic arch [1,2,3] and is key to achieving successful outcomes [1,2,3,4,5,6,7].

The technical factors affecting perfusion, such as the perfusion site, perfusate temperature, and flow rate, have not been definitively established [2,5]. The most frequently utilized cannulation sites include the ascending aorta or a peripheral artery such as the femoral, subclavian, axillary, or carotid artery [2,8,9,10,11,12], presenting potential advantages and disadvantages [2,3].

Cannulation of the innominate artery (IA) is a less common approach [12,13,14]. Banbury and Cosgrove initially documented IA cannulation in 2000 [15]. The cannulation technique may be direct or involve a side graft [16]. In capable hands, direct cannulation proves to be a feasible and expeditious procedure, facilitating secure systemic perfusion and the administration of ACP as necessary. This approach offers the advantage of avoiding aortic manipulation, reducing the risk of stroke associated with embolization of atheromatous debris or thrombus from the cannulation site. It also minimizes the need for a second incision as the cannulation is performed after the sternotomy and it ensures the antegrade cerebral perfusion. This method involves delivering oxygenated blood directly to the brain through the carotid arteries during periods of circulatory arrest, which is necessary for the repair of the aorta, thereby serving as an additional measure to mitigate the risk of intraoperative malperfusion. In this regard, it is an excellent alternative to standard artery cannulation sites.

## 2. Materials and Methods

This was a retrospective, single-institutional-cohort study approved by the partners’ Institutional Review Board. The need for patient consent was waived due to the retrospective nature of this study. Pertinent data are included in the manuscript and the Appendix A.

### 2.1. Study Design and Variables

The primary outcome measure analyzed was operative mortality, with the secondary outcome measure being postoperative neurologic events.

Operative mortality was specifically defined as any death occurring within 30 days after an operation or before hospital discharge throughout a five-year follow-up period.

Additionally, neurological deficits were categorized as either permanent (PND) or transient (TND) based on the duration of symptoms at discharge. All patients presenting with neurological deficits underwent evaluation with a CT scan [17].

The operative times were delineated as follows: myocardial ischemia time referred to the duration from the point of aorta cross-clamping to clamp removal, cardiopulmonary bypass time (CPB) denoted the period when the patient was sustained by cardiopulmonary bypass, and antegrade cerebral perfusion (ACP) time referred to the duration of circulatory arrest during which the patient received ACP.

### 2.2. Statistical Analysis

Quantitative data were presented as the median and interquartile range (IQR), while categorical variables were shown as quantitative data, specifically counts and percentages. Associations between categorical variables were examined using either the Pearson Chi-square test or Fisher’s exact test. Two-group comparisons for quantitative variables were evaluated using the Mann–Whitney test, and the Kruskal–Wallis test was used for multiple-group comparisons. A two-sided *p*-value of less than 0.05 was considered to indicate statistical significance. Categorical variable associations were examined using either the Pearson Chi-square test or Fisher’s exact test. Comparisons between two groups for quantitative variables were evaluated using the Mann–Whitney test, and the Kruskal–Wallis test was utilized for multiple-group comparisons. A two-sided *p*-value of less than 0.05 was deemed statistically significant. The odds ratio (OR) and the corresponding 95% confidence interval were calculated using logistic regression analysis. Five-year overall survival was determined using the Kaplan–Meier method, with survival intervals computed from the date of intervention to either death or the last follow-up at 5 years. This analysis was based on 130 out of 208 patients to ensure adherence to a five-year follow-up period. All statistical analyses were performed using SPSS-IBM software, version 25.

### 2.3. Surgical Technique

#### Cannulation

When dealing with the presence of an intimal flap or extensive calcification of the innominate artery (IA), direct cannulation poses challenges. Before pericardiectomy, it is important to identify and retract the innominate vein using an umbilical tape to aid in visualizing the IA. The pericardium is opened only for elective procedures before the surgical exposure of the IA. Typically, the IA is dissected and exposed up to the bifurcation, then encircled with umbilical tape. The left carotid and left subclavian arteries are similarly prepared using the same technique. After a thorough assessment, two simple box-shaped 5-0 polypropylene stitches, positioned side by side in opposite directions and reinforced with Teflon-felt pledgets, are placed on the anterior wall of the IA to create a space for cross-clamping (Figure 1). This technique minimizes arterial deformation after the stitches are tied. Selecting the appropriate cannula size is crucial to ensure adequate flow. The cannula tip should be designed straight with an adjustable ring to prevent over-insertion and potential obstruction of flow to the head vessels. The correct angulation of the cannula tip for orthogonal positioning should be determined based on the size of the vessel, as demonstrated in video 1. Throughout the procedure, systemic blood pressure was continuously monitored using the right and left peripheral arterial lines. The compatibility between the innominate artery and cannula diameter was deemed satisfactory. Following the introduction of the cannula, we keep the tip inside the vessel for 5 mm, then the purse strings are tightened, and the cannula is secured to the tourniquets (Figure 2).

Once venous cannulation is completed, cardiopulmonary bypass is initiated. In the event of a necessary circulatory arrest, the patient is cooled to 26°/28° C. The INVOS Cerebral/Somatic Oximeter (Somanetics, Troy, MI, USA) and the Bispectral Index (Masimo, Irvine, CA, USA) were used to continuously monitor cerebral perfusion during the surgical treatment of the aortic arch, which involved planned interruptions of normal brain perfusion. Tourniquets were applied around the proximal IA and left carotid artery. Upon reaching the target temperature and initiating systemic circulatory arrest, we administered unilateral antegrade selective cerebral perfusion (SCP) at a flow rate of 10–15 mL/kg/min. To maintain mean blood pressure at 50 to 70 mmHg during systemic circulatory arrest, we monitored via the right radial arterial line. In cases where there is lateral asymmetry on the EEG and/or a >50% decrease in cerebral oximetry by NIRS, we should consider initiating bilateral aSCP through the left carotid artery perfusion. It is worth noting that preoperative cranial CT angiography is not recommended as a standard procedure before arch surgery, as the anatomical incompleteness of the circle of Willis does not necessarily correlate with insufficient cerebral protection during hypothermic unilateral cerebral perfusion [18]. Although the circle of Willis is likely not the exclusive pathway for cerebral cross-perfusion, extracranial collateral circulation, especially in patients with an incomplete circle of Willis, seems to be significant [19,20].

When discontinuing the cardiopulmonary bypass, the IA cannula is removed by securely tying down the purse strings and monitoring the right radial or brachial arterial blood pressure.

## 3. Results

Between January 2010 and December 2021, 208 patients underwent direct intra-arterial cannulation for thoracic aorta surgery. The group consisted of 133 (63.9%) male and 75 (36.1%) female patients, with a median age of 69 (59–76). Among them, 60 patients (28.8%) required emergency treatment for type A aortic dissection. The population’s characteristics are summarized in Table 1. Details of the surgeries, including proximal aortic reconstruction and concurrent procedures, are outlined in Figure 3. There were no reported vascular injuries associated with the inflow cannulation procedure. There was no necessity to change the cannulation site in any of the patients. Hypothermic circulatory arrest with ACP was utilized in all cases that necessitated open distal anastomosis (*n* = 110 patients; 52.9%). Among the 110 patients, 34 (30.9%) underwent bilateral ACP. The median duration of cardiopulmonary bypass was 140 min [108–184], of aortic cross-clamping was 97 min [77–138], and of antegrade cerebral perfusion was 31.5 min [22–56] (Table 2). Postoperative complications are detailed in Table 3. Infections were the most represented postoperative complications (23 patients, 11%), followed by neurological events (11 patients, 8.2%) and renal failure (15 patients, 7.2%).

### 3.1. Mortality

The operative mortality rate was 5.3%, as shown in Table 4. Following emergency intervention for type A aortic dissection, 10% of patients (six individuals) passed away, while the mortality rate in the elective surgery group was 3.4% (five patients). This is consistent with the logistic EuroSCORE findings of 30.1% [19.71–59.11] versus 13.9% [8.4–23.9] (*p* < 0.001). Among the deceased, five deaths resulted from ventricular failure, four of whom had undergone surgery for type A aortic dissection involving coronary vessels and experienced impaired ventricular function due to extensive myocardial infarction. Additionally, a patient underwent mitral-aortic valve replacement along with complete aortic arch replacement and reimplantation of the supra-aortic trunks. Cardiopulmonary bypass (CPB) and cross-clamping times were approximately 322 and 188 min, respectively, while hypothermic circulatory arrest with antegrade cerebral perfusion (ACP) lasted approximately 37 min.

During this study, one patient died from sepsis 38 days after the operation. This individual had previously undergone reoperation for an aortic arch pseudoaneurysm following a Bentall procedure for type A acute aortic dissection. Additionally, five patients died from neurological complications: two had ischemic brain lesions on CT scans, one had both ischemic and hemorrhagic lesions, and another patient was diagnosed with sagittal sinus thrombosis. On the 16th day after the operation, one patient died from irreversible neurological damage without undergoing a CT scan for neurological lesions, suggesting a potential brainstem injury. The overall long-term mortality rate at five years was 7.7%, with a total of 16 patient fatalities. Notably, the five-year mean survival for patients who underwent surgery for dissection, calculated for 130 subjects, was found to be similar to those who had elective surgery at 4.4 years versus 4.6 years, respectively, with a *p*-value of 0.46 (Figure 4).

### 3.2. Neurological Events

Seventy patients (8.2%) experienced neurological events, with eight of them (3.8%) suffering from permanent neurological deficits. Among the eleven patients with negative brain CT scans, nine had transient neurological deficits with complete functional recovery upon discharge, while two suffered from permanent neurological deficits. Within the group with permanent neurological deficits, six patients (66.7%) passed away (Figure 5).

## 4. Discussion

Before thoracic aorta surgery, it is crucial to perform a comprehensive preoperative assessment. This should include a thorough clinical examination, detailed imaging evaluation, and careful surgical planning for both elective and emergencies, as outlined in scholarly sources [1,2,3]. The surgical approach should be tailored based on the specific pathology, anatomy, and experience of the surgical team [2]. Adequate management of perfusion is essential to support the surgical repair process [1,2,3].

The selected cannulation strategy should ensure sufficient perfusion flow and optimal antegrade perfusion. It should also provide the flexibility to deliver antegrade brain perfusion. The decision-making process for the arterial cannulation site is of significant importance. In recent years, various sites have been suggested for cannulation, each demonstrating both advantages and disadvantages [2]. Direct aortic cannulation involves placing the cannula directly into the ascending aorta. This method provides physiological antegrade perfusion, which can be advantageous in maintaining stable hemodynamics and reducing complications associated with retrograde flow and can be instituted rapidly. However, direct cannulation of the dissected ascending aorta can lead to rupture, especially if the aortic wall is fragile or thin. The position of the inserted cannula may interfere with the proximal anastomosis. The procedure requires precise placement of the cannula into the true lumen, often necessitating imaging guidance such as transesophageal echocardiography or epiaortic ultrasonography to confirm correct positioning. Despite these potential complications, direct aortic cannulation remains a viable option due to its advantages in providing antegrade perfusion and reducing the risk of retrograde flow complications seen with femoral cannulation.

Despite this, there is a growing body of evidence that supports the effectiveness of axillary/subclavian or IA cannulation for providing antegrade body and brain perfusion [7,8,9,10,11,12,13,14,15]. Although the initial description of IA cannulation can be traced back to the early 1900s [16], recent years have seen a surge in its utilization due to increasing experience in this area [12,13,14]. A thorough understanding of anatomic structures and surgical techniques has been pivotal in this trend.

Exposure can be quickly and easily achieved through intra-arterial (IA) approaches, making them well-suited for a variety of clinical situations. Direct IA cannulation eliminates the need for a secondary surgical incision, providing a simple and efficient method while also typically ensuring an optimal blood flow rate due to its larger size [14,15,16,17,18,19,20,21,22]. Studies conducted by Chu et al. [22] and Di Eusanio et al. [13] have demonstrated that innominate artery cannulation is a feasible and safe alternative with minimal requirement for a side graft in most cases.

Moreover, Garg et al. demonstrated the viability and safety of utilizing the innominate artery for antegrade cerebral perfusion (ACP) and hypothermic circulatory arrest (HCA) during aortic arch reconstruction [23]. In the initial randomized trial focusing on elective repair of the ascending aorta and proximal arch surgery, Peterson et al. determined that innominate artery cannulation results in comparable neurological outcomes to axillary cannulation [24].

However, the trial’s focus on elective surgical procedures raises the question of outcomes in emergent surgeries, such as aortic dissection, which remains unanswered. The outcome of emergent aortic dissection, particularly acute type A aortic dissection (ATAAD), is highly dependent on timely surgical intervention. The American College of Cardiology and the American Heart Association recommend urgent surgical evaluation for suspected or diagnosed acute type A aortic dissection due to the significantly higher mortality rate associated with medical management alone.

Surgical intervention has been shown to reduce the immediate risk of fatal complications such as aortic rupture, cardiac tamponade, and myocardial ischemia. Data from the International Registry of Acute Aortic Dissection (IRAD) indicate that surgical mortality rates have decreased from 25% to 18% over recent decades, while medical management alone has a mortality rate of approximately 57%. Despite the improvements in surgical outcomes, the overall prognosis remains guarded. A study evaluating long-term outcomes reported a 10-year survival rate of approximately 60–65% following surgical repair of ATAAD. Factors such as preoperative shock, tamponade, and malperfusion syndromes significantly increase the risk of mortality and complications [25]. Preventza et al. reported favorable outcomes in a small cohort of patients with aortic dissection [7]. The utilization of intra-arterial (IA) cannulation is linked to very rare contraindications compared to other sites. Limitations of its application include the involvement of aortic dissection and the presence of atheromatous plaques. Redo surgery, obesity, or a hostile chest are generally not considered contraindications. If IA cannulation is not feasible, the preferred sites are the subclavian artery or echo-guided ascending aorta, while femoral artery cannulation is considered a last resort.

Our analysis revealed that the mortality rate for patients undergoing operative interventions was 5.3%. Specifically, 10% of those who received emergency intervention for type A aortic dissection did not survive, while the mortality rate for elective surgery was 3.4%. Despite the difference in mortality rates being only slightly above the threshold for statistical significance as predicted by the logistic EuroSCORE, it was apparent that the risk of death during emergency surgery was significantly higher. This mortality rate was consistent with findings from other researchers and was not found to be linked to the arterial cannulation site.

In the context of neurological events (as shown in Table 3), aortic dissection is highly linked to postoperative neurological events (20% vs. 3.4%, *p* < 0.001).

The data show that approximately 30% of cases involved dissections, which led to an overestimation of events at 8.2%. After excluding patients with dissections, neurological events occur at 3.4% (5 out of 148 patients), which is consistent with the existing scientific literature [26]. Our findings reveal an 11.7-fold increase in the risk of postoperative neurological events associated with a positive history of previous cerebrovascular events. Notably, among patients undergoing elective cardiac surgery (excluding the dissection factor), those with preoperative neurological events experienced a roughly 25% rate of neurological complications, compared to 2.8% of patients without these risk factors. This difference approached statistical significance with an odds ratio of 11.667 and a 95% confidence interval of 0.985 to 138.176 (*p* = 0.051).

Our findings indicate that, except for emergency procedures, there is no significant difference in neurological risk for patients undergoing circulatory arrest with unilateral or bilateral perfusion (*p* > 0.05).

The neurological risk remains consistent when comparing patients undergoing elective cardiac surgery, regardless of whether circulatory arrest is involved (*p* > 0.05). Left untreated, patients with aneurysm and dissection have a limited life expectancy. The literature reports a 57% 5-year mortality rate for the spontaneous progression of chronic dissection and aneurysm. Immediate surgical intervention is necessary in cases of acute aortic dissection due to the extremely high mortality rates following disease onset [11,12].

As competence in traditional cannulation methods continues to advance, it is increasingly important to carefully weigh the advantages and drawbacks of each approach to select the most suitable site and technique for cannulation. Surgical decision-making should prioritize speed, dependability, and patient safety.

The direct IA cannulation technique represents an efficient and expedient option. Regrettably, one of its primary drawbacks is its lack of widespread adoption. Nevertheless, based on our experience, achieving access via a sternotomy enables clear visualization of the vessel and accurate placement of the cannula compared to the axillary approach. Moreover, sternotomy allows for rapid control of complications such as cardiac tamponade.

Once inserted, the innominate cannula must extend only slightly into the vessel, thereby preventing any flow obstruction to the systemic vessels.

In our experience, direct cannulation has been associated with minimal surgical-related complications. Eldeiry et al. have shown that it leads to better control of the surgical field, resulting in reduced blood loss and a significantly lower need for transfusions of red blood cells, platelets, and fresh frozen plasma [27].

Cerebral oximetry can provide an early warning of potential brain injury during surgery, prompting the surgical team to take necessary measures such as adjusting the arterial cannula position or choosing a different cannulation site. In cases of circulatory arrest, when cerebral perfusion is not equally supported with unilateral selective antegrade cerebral perfusion (ACP), differences in hemispheric regional oxygen saturation (rSO2) could indicate inadequate perfusion in the left hemisphere, signaling the need to establish bilateral brain perfusion promptly [28,29,30]. In our surgical procedures for the thoracic aorta, we prefer direct IA cannulation. This technique is straightforward, efficient, avoids the complications of a secondary incision, and has the potential to enhance surgical outcomes while safely achieving ACP [31]. Contraindications include dissection and atherosclerotic lesions.

### Limitations

This study has some limitations, including its observational design rather than a comparative one and the fact that it is based on a single-center trial. Although innominate artery cannulation represents a valid alternative to axillary cannulation, further comparative studies are needed.

## 5. Conclusions

Various cannulation strategies come with their own set of advantages and drawbacks in cardiopulmonary bypass procedures.

At our institution, the approach to proximal thoracic aortic surgery has progressed to more frequent use of IA cannulation in both elective and urgent procedures as a safe and effective means of providing arterial inflow during cardiopulmonary bypass. This shift is particularly beneficial for complex aortic arch procedures, as it simplifies the surgical technique, promotes safety through expertise, and has the potential to reduce the surgical time. By aligning with the natural flow of blood, this approach minimizes malperfusion complications during CPB and enables anterograde cerebral perfusion during circulatory arrest, ultimately leading to favorable short- and long-term outcomes. Although direct cannulation of IA is an excellent alternative to standard artery cannulation sites, it necessitates a thorough assessment of anatomy, systematic preoperative imaging review, and collaborative efforts among surgeons, anesthesiologists, and perfusionists to optimize perfusion management.

## Figures and Tables

**Figure 1 jcm-14-02684-f001:**
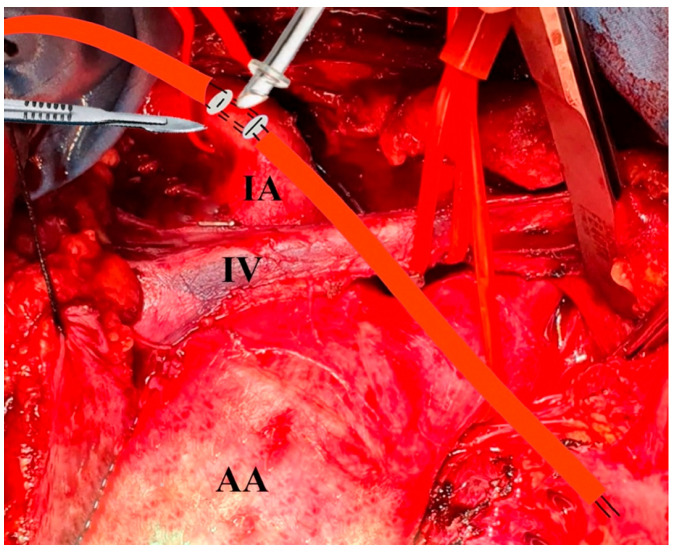
Innominate artery direct cannulation. AA: ascending aorta; IV: innominate vein; IA: innominate artery.

**Figure 2 jcm-14-02684-f002:**
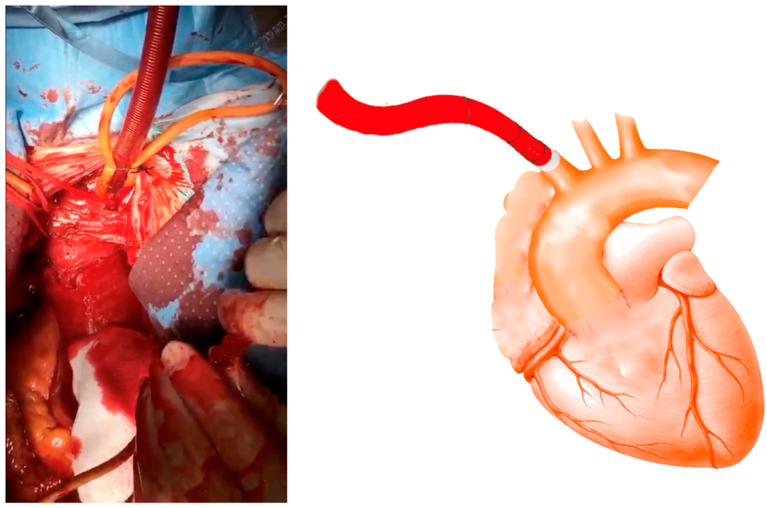
Schematic and final aspect of direct IA cannulation.

**Figure 3 jcm-14-02684-f003:**
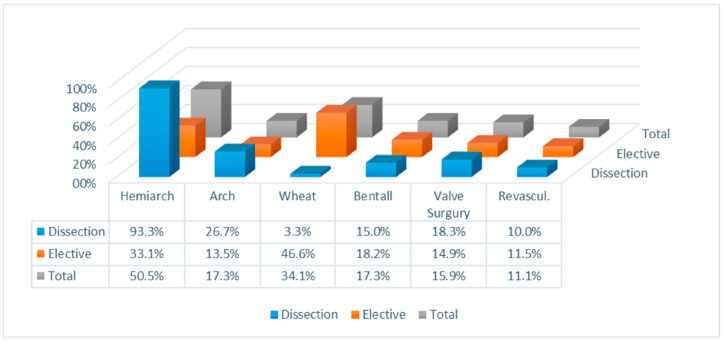
Surgical procedures.

**Figure 4 jcm-14-02684-f004:**
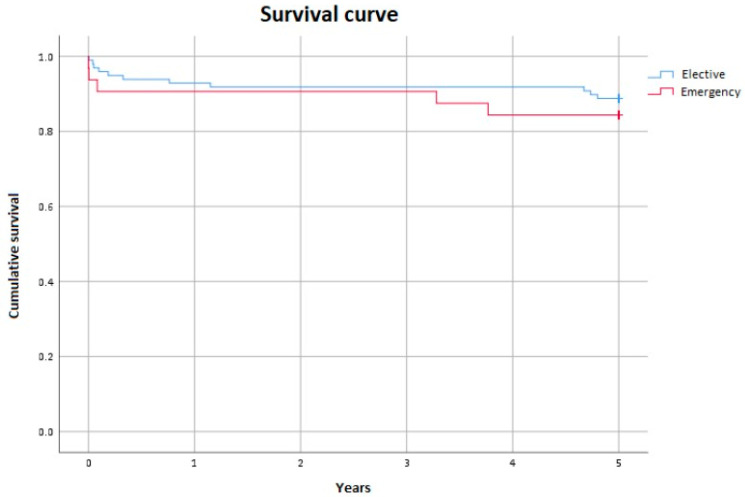
Survival curve on follow-up.

**Figure 5 jcm-14-02684-f005:**
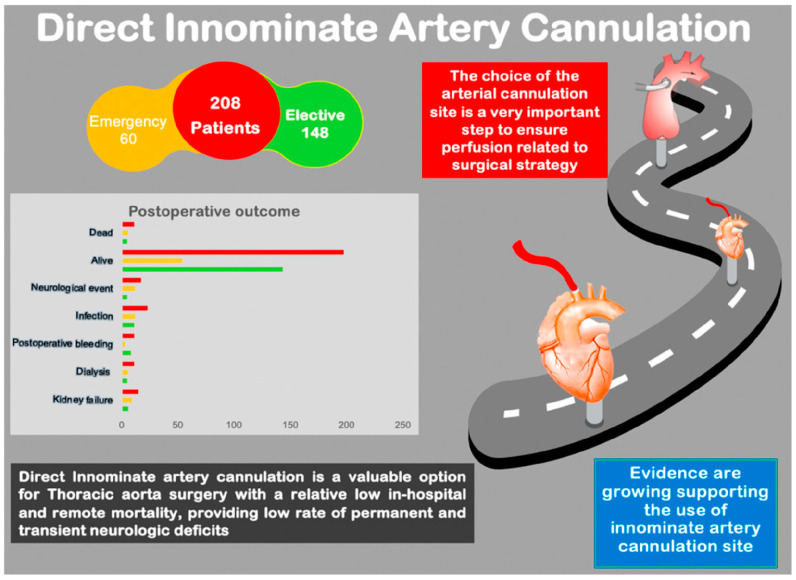
Direct innominate artery cannulation and outcome.

**Table 1 jcm-14-02684-t001:** Patient demographics and anamnestic/clinical information (*n* = 208).

	*n*	%
Male	133	63.9
Female	75	36.1
Median Age [IQR]	69.00	[59.25–76]
Smoking	91	43.8
Hypertension	173	83.2
Diabetes Mellitus	12	5.8
Dysilipidemia	73	35.1
COPD	21	10.1
Peripheral Vascular Disease	65	31.3
History of Myocardial Infarction	3	1.4
Renal Failure	19	9.1
Median EF% [IQR]		59 [55–60]
EuroSCORE % [IQR]		4.04 [2.27–8.93]

Legend: IQR interquartile range, COPD chronic obstructive pulmonary disease, EF ejection fraction.

**Table 2 jcm-14-02684-t002:** Operative times.

	Median [IQR]
CPB	140.50 [107.75–184]
Aortic Cross-Clamping	97 [77–138.25]
ACP (*n* = 110)	31.50 [22–56.25]

Legend: CPB cardiopulmonary bypass, ACP antegrade cerebral perfusion.

**Table 3 jcm-14-02684-t003:** Postoperative complications.

	Total	Elective	Dissection Type A
Patients	208	148	60
Kidney Failure	7.2%	4.1%	15%
	15	6	9
Dialysis	5.3%	3.4%	10%
	11	5	6
Postoperative Bleeding	5.3%	5.4%	5%
	11	8	3
Infections	11.1%	7.4%	20%
	23	11	12
Neurological Events	8.2%	3.4%	20%
(PND and TND)	17	5	12

Legend: PND permanent neurological dysfunction, TND transient neurological dysfunction.

**Table 4 jcm-14-02684-t004:** Operative death.

	Total	Elective	Dissection Type A
Patients	208	148	60
Alive	94.7%	96.6%	90%
	197	143	54
Dead	5.3%	3.4%	10%
	11	5	6
Total	100%	100%	100%

## Data Availability

The data that support the findings of this study are available from the corresponding author, [C.C], upon reasonable request.

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
