# Peer review of "Direct Innominate Artery Cannulation for Thoracic Aortic Surgery"

_jcm, 2025, doi:10.3390/jcm14082684_

Round 1
Reviewer 1 Report
Comments and Suggestions for Authors
Dear authors,
It was a pleasure reviewing your manuscript titled "Direct innominate artery cannulation for thoracic aorta elective 2 or emergent surgery". I congratulate the authors for achieving good outcomes of thoracic aortic surgery using Innominate artery cannulation. The authors have described their surgical outcomes with a help of retrospective observational study.
I have certain recommendation for the manuscript as mentioned below
Title: I recommend shorter title: Direct innominate artery cannulation for thoracic aortic surgery.
Abstract: I recommend balancing the Methods and Results section. The method section ideally should read like “A single center retrospective study of all cases that underwent thoracic aortic surgery between ___were studied”. Primary outcome was __ and secondary outcome was __. In the results all the types of aortic operations are redundant.
Materials and methods: The language needs to be more standard rather than saying “The
surgical team extracted the data from a meticulously managed clinical database.” The more standard wordings are as mentioned for the abstract “A single center retrospective study..”
“All participants provided informed consent for data collection.” Participants usually cannot consent for a retrospective study. Please confirm if the study was retrospective and you reached out to all participants for ‘consent’. The consent usually waived by ethics committees for retrospective studies.
Definitions in 82-86 are standard and thus redundant.
Results: It is customary to include percentages while mentioning number of patients, example: the number of male patients were 133(63.9%). All median values should have interquartile range in parenthesis.
Surgical technique: should be written before the results. It would be good to know how many millimeters actually stays in the innominate artery. There is a one centimeter mark in most cannulae. Do go past that mark or that mark stays outside the Innominate artery
Lines 201 onwards in surgical technique actually belongs the results section.
The cardiopulmonary bypass times, cross clamps time and antegrade cerebral perfusion times should be mentioned in the text as well with median values and interquartile range in parenthesis.
The main complications should also be mentioned in the text. Line 210 is redundant, p-value should be mentioned as is. By convention p>0.05 is not significant difference and should be mentioned in the statistical section. Line 228-230: The long-term or intermediate term mortality is more conventional than remote mortality rate. Lines 237- 240 is a repetition of 224-226.
Discussion: Lines 338-341 need a citation.
Lines 359-361: The table 4 is concerned with death and not neurological dysfunction. Table 3 shows neurological dysfunction is more prevalent in aortic dissection. Whether it is significant or not is not mentioned in the table since p value is not mentioned. Even if p-value was significant, aortic dissection cannot be assumed to ‘predictor’. In order to predict an event, a univariate or multivariable analyses should be undertaken.
Mentioning p>0.99 is not the norm rather a p>0.05 is the norm for insignificant difference.
Limitations is supposed to different paragraph with title. I think the major limitation is that innominate artery cannulation is not compared with any other technique.
Comments on the Quality of English Language
Dear authors,
The use of English language is ok but I have mentioned above wherein some non-standard phrases are used and would request you either change it or take a professional language help.
Author Response
Response to reviewer 1
Thank you very much for taking the time to review this manuscript. Please find the detailed responses below and the corresponding revisions/corrections highlighted/in track changes in the re-submitted files
- Title: I recommend shorter title: Direct innominate artery cannulation for thoracic aortic surgery.
Thank you for your comment. We inserted it in the title
Page 1 line 2
- Abstract: I recommend balancing the Methods and Results section. The method section ideally should read like “A single center retrospective study of all cases that underwent thoracic aortic surgery between ___were studied”. Primary outcome was __ and secondary outcome was __
- Definitions in 82-86 are standard and thus redundant.
Agree. We have removed them form the text
A single center retrospective study of 208 cases that underwent thoracic aortic surgery between January 2010 and December 2021, were studied. Primary outcome were in-hospital and remote mortality and secondary outcome were adverse neurologic events.
Page 2 line 16-19
- Results: It is customary to include percentages while mentioning number of patients, example: the number of male patients were 133(63.9%). All median values should have interquartile range in parenthesis.
Thank you, we added them in the results.
The group consisted of 133 (63.9%) male and 75 (36.1%) female patients, with a median age of 69 (59-76).
Page 5 line 208
- Surgical technique: should be written before the results. It would be good to know how many millimeters actually stays in the innominate artery. There is a one centimeter mark in most cannulae. Do go past that mark or that mark stays outside the Innominate artery
We change the position of the surgical technique in the text as suggested.
We modified the text as follow: Following the introduction of the cannula, we keep the tip inside the vessel for 5 mm, then the purse strings are tightened, and the cannula is secured to the tourniquets (Fig.3).
Page 4 line 177
- Lines 201 onwards in surgical technique actually belongs the results section.
Thank you for your kind suggestion, we changed the text
Page 4 line 162
- The cardiopulmonary bypass times, cross clamps time and antegrade cerebral perfusion times should be mentioned in the text as well with median values and interquartile range in parenthesis
Thank you for your suggestion. We agree and modified the text.
The median Cardio Pulmonary Bypass was of 140[108-184] minutes, aortic cross-clamping was of 97 [77-138] and Antegrade Cerebral Perfusion time was 31.5[22-56] minutes (Table 2).
Page 5 line 216
- The main complications should also be mentioned in the text
We agree and we added them
Infections were the most represented postoperative complications (23 patients, 11%), followed by neurological events (11 patients, 8.2%) and renal failure (15 patients, 7.2%).
Page 5 line 218
- Line 210 is redundant, p-value should be mentioned as is. By convention p>0.05 is not significant difference and should be mentioned in the statistical section.
Agree. We removed it
- Line 228-230: The long-term or intermediate term mortality is more conventional than remote mortality rate.
Agree
We changed into long-term mortality
- Lines 237- 240 is a repetition of 224-226.
Agree. We removed it from the text
- Discussion: Lines 338-341 need a citation.
We added the reference and modified the bibliography25. Peterson MD, Garg V, Mazer CD, Chu MWA, Bozinovski J, Dagenais F, MacArthur RGG, Ouzounian M, Quan A, Jüni P, Bhatt DL, Marotta TR, Dickson J, Teoh H, Zuo F, Smith EE, Verma S; ACE CardioLink-3 Trial Working Group. A randomized trial comparing axillary versus innominate artery cannulation for aortic arch surgery. J Thorac Cardiovasc Surg. 2022 Nov;164(5):1426-1438.e2. doi: 10.1016/j.jtcvs.2020.10.152. Epub 2020 Dec 1.
Page 15 line 712
- Lines 359-361: The table 4 is concerned with death and not neurological dysfunction. Table 3 shows neurological dysfunction is more prevalent in aortic dissection. Whether it is significant or not is not mentioned in the table since p value is not mentioned. Even if p-value was significant, aortic dissection cannot be assumed to ‘predictor’. In order to predict an event, a univariate or multivariable analyses should be undertaken.
Thank you for your suggestion. We modified the text as follow
In the context of neurological events (as shown in Table 3), aortic dissection is highly linked to postoperative neurological events (20% vs 3.4%, p < 0.001).
Page 12 Line 517
- Mentioning p>0.99 is not the norm rather a p>0.05 is the norm for insignificant difference.
Thank you for your suggestion, we modified it.
- Limitations is supposed to different paragraph with title. I think the major limitation is that innominate artery cannulation is not compared with any other technique
Agree, we changed the text
Although Innominate artery cannulation represents a valid alternative to axillary cannulation, however, further comparative studies are needed.
Page 13 Line 583
Reviewer 2 Report
Comments and Suggestions for Authors
JCM-25-3458809
Direct innominate artery cannulation for thoracic aorta elective 2 or emergent surgery
In this paper the authors present the clinical outcomes of patients who underwent direct innominate cannulation for arterial inflow during thoracic aorta surgery.
Comments
The paper brings globally interesting data which can be useful for the clinic. A few points should however be completed and modified:
-“…The most frequently utilized cannulation sites include the ascending aorta…..” pros and cons of each approach should detailed, in order to bring out later the advantages of the innominate cannulation. This would better motivate the work done
-“…This approach offers the advantage of avoiding aortic manipulation, minimizing the need for a second incision, reducing the risk of retrograde cerebral embolism, and accommodating a more suitable size…” …not evident for non-specialists, this should be more detailed (maybe with a schema comparing the approaches)
-“…However, the trial's focus on elective surgical procedures raises the question of outcomes in emergent surgeries, such as aortic dissection, which remains unanswered…” this point could be justified in more details. What is the key concern of emerging surgeries...this should be deepened for a larger journal audience purpose
-“…Nevertheless, based on our experience, achieving access via a sternotomy enables…” isn't this the case when a subclavian or axillary approach is considered ? Again, maybe a more detailed description and comparison in the introduction of the different approaches could help to better understand the advantage of the approach
Author Response
Response to reviewer 2
Thank you very much for taking the time to review this manuscript. Please find the detailed responses below and the corresponding revisions/corrections highlighted/in track changes in the re-submitted files
- The most frequently utilized cannulation sites include the ascending aorta…..” pros and cons of each approach should detailed, in order to bring out later the advantages of the innominate cannulation. This would better motivate the work done
Thank you for pointing this out. We agree with this comment, therefore, we have changed the text as follow.
Page 11 line 470
Direct aortic cannulation involves placing the cannula directly into the ascending aorta. This method provides physiological antegrade perfusion, which can be advantageous in maintaining stable hemodynamics and reducing complications associated with retrograde flow and can be instituted rapidly. However, direct cannulation of the dissected ascending aorta can lead to rupture, especially if the aortic wall is fragile or thin. The position of the inserted cannula may interfere with the proximal anastomosis. The procedure requires precise placement of the cannula into the true lumen, often necessitating imaging guidance such as transesophageal echocardiography or epiaortic ultrasonography to confirm correct positioning. Despite these potential complications, direct aortic cannulation remains a viable option due to its advantages in providing antegrade perfusion and reducing the risk of retrograde flow complications seen with femoral cannulation
- This approach offers the advantage of avoiding aortic manipulation, minimizing the need for a second incision, reducing the risk of retrograde cerebral embolism, and accommodating a more suitable size…” …not evident for non-specialists, this should be more detailed (maybe with a schema comparing the approaches)
Agree. I have modified the text to emphasize that point. Page 2 line 68-74
This approach offers the advantage of avoiding aortic manipulation reducing the risk of stroke associated with embolization of atheromatous debris or thrombus from the cannulation site. It also minimizes the need for a second incision as the cannulation is performed after the sternotomy and it ensure the antegrade cerebral perfusion. This method involves delivering oxygenated blood directly to the brain through the carotid arteries during periods of circulatory arrest, which is necessary for the repair of the aorta, thereby serving as an additional measure to mitigate the risk of intraoperative malperfusion.
- However, the trial's focus on elective surgical procedures raises the question of outcomes in emergent surgeries, such as aortic dissection, which remains unanswered…” this point could be justified in more details. What is the key concern of emerging surgeries...this should be deepened for a larger journal audience purpose
We agree with this comment, therefore we have changed the text as follow
Page 10 line 503-518
However, the trial's focus on elective surgical procedures raises the question of outcomes in emergent surgeries, such as aortic dissection, which remains unanswered. The outcome of emergent aortic dissection, particularly acute type A aortic dissection (ATAAD), is highly dependent on timely surgical intervention. The American College of Cardiology and the American Heart Association recommend urgent surgical evaluation for suspected or diagnosed acute type A aortic dissection due to the significantly higher mortality rate associated with medical management alone.
Surgical intervention has been shown to reduce the immediate risk of fatal complications such as aortic rupture, cardiac tamponade, and myocardial ischemia. Data from the International Registry of Acute Aortic Dissection (IRAD) indicate that surgical mortality rates have decreased from 25% to 18% over recent decades, while medical management alone has a mortality rate of approximately 57%
Despite the improvements in surgical outcomes, the overall prognosis remains guarded. A study evaluating long-term outcomes reported a 10-year survival rate of approximately 60-65% following surgical repair of ATAAD. Factors such as preoperative shock, tamponade, and malperfusion syndromes significantly increase the risk of mortality and complications. [26]
- Nevertheless, based on our experience, achieving access via a sternotomy enables…” isn't this the case when a subclavian or axillary approach is considered? Again, maybe a more detailed description and comparison in the introduction of the different approaches could help to better understand the advantage of the approach
We agree with this comment, we revised the text.
Page 12 line 562-565
Nevertheless, based on our experience, achieving access via a sternotomy enables clear visualization of the vessel and accurate placement of the cannula compared to the axillary approach. Moreover, sternotomy allows for rapid control of complications such as cardiac tamponade.
Round 2
Reviewer 2 Report
Comments and Suggestions for Authors
thanks for having addressed my concerns